# Corrosion Resistance of Al_0.5_CoCrFeNiCu*_x_*Ag*_y_* (*x* = 0.25, 0.5; *y* = 0, 0.1) High-Entropy Alloys in 0.5M H_2_SO_4_ Solution

**DOI:** 10.3390/ma16093585

**Published:** 2023-05-07

**Authors:** Olga Samoilova, Svetlana Pratskova, Nataliya Shaburova, Ahmad Ostovari Moghaddam, Evgeny Trofimov

**Affiliations:** 1Department of Materials Science, Physical and Chemical Properties of Materials, South Ural State University, 76 Lenin Av., 454080 Chelyabinsk, Russia or samoylova_o@mail.ru (O.S.); or ostovarim@susu.ru (A.O.M.); tea7510@gmail.com (E.T.); 2Research & Innovation Services, South Ural State University, 76 Lenin Av., 454080 Chelyabinsk, Russia; se_pratskova@mail.ru; 3Department of Analytical and Physical Chemistry, Chelyabinsk State University, 129 BratievKashirinyh Street, 454001 Chelyabinsk, Russia

**Keywords:** high-entropy alloys, corrosion resistance, polarization, electrochemical behavior

## Abstract

The electrochemical behavior of the as-cast Al_0.5_CoCrFeNiCu*_x_*Ag*_y_* (*x* = 0.25, 0.5; *y* = 0, 0.1) high-entropy alloys (HEAs) in a 0.5M H_2_SO_4_ solution was studied. Polarization measurements were carried out in a standard three-electrode electrochemical cell at room temperature using a platinum counter electrode and a saturated silver chloride reference electrode. For Al_0.5_CoCrFeNiCu_0.5_ and Al_0.5_CoCrFeNiCu_0.5_Ag_0.1_, copper segregation along the grain boundaries was observed, which highly dissolved in the sulfuric acid solution and resulted in low corrosion resistance of the samples. Introducing Ag into Al_0.5_CoCrFeNiCu_0.25_ HEA led to the precipitation of a copper–silver eutectic structure, in which the copper regions were selectively dissolved in the sulfuric acid solution. Al_0.5_CoCrFeNiCu_0.25_ exhibited the best corrosion resistance with the corrosion current density of *I*_corr_ = 3.52 ± 0.02 μA/cm^2^, significantly superior to that of the Al_0.5_CoCrFeNi sample without copper and silver (*I*_corr_ = 6.05 ± 0.05 μA/cm^2^). Finally, the results indicated that suppressing elemental segregation by annealing or tailoring chemical composition is essential to improve the corrosion resistance of Al_0.5_CoCrFeNiCu*_x_*Ag*_y_* HEAs.

## 1. Introduction

The concept of high-entropy alloys (HEAs), proposed in 2004 by Cantor and Yeh [1,2], has attracted significant scientific interest owing to their unique properties such as high strength and ductility [3,4], good fatigue resistance [5,6], and improved tribological characteristics [7,8,9]. Considering the relatively large number of materials working under extreme conditions, the behavior of HEAs at both ultralow [10,11,12] and elevated temperatures [13,14,15], as well as their corrosion resistance in various aggressive solutions [16,17], have been also investigated.

According to the literature [16,17,18,19,20], Al*_x_*CoCrFeNi high-entropy alloys are among the most promising corrosion-resistant materials. Li et al. [18] reported that AlCoCrFeNi exhibits higher corrosion resistance compared to steel AISI 1045 in 3.5 wt.% NaCl solution. It was shown that the corrosion current of this HEA (9.3 × 10^−8^ A/cm^2^) was three orders of magnitude lower than that of steel (1.17 × 10^−5^ A/cm^2^) [18]. Shi et al. [19] showed that Al_0.3_CoCrFeNi and Al_0.5_CoCrFeNi HEAs with a reduced aluminum concentration, exhibited very low corrosion current density values of 8.35 × 10^−8^ A/cm^2^ and 2.52 × 10^−7^ A/cm^2^ in 3.5 wt.% NaCl solution, respectively. It has been further revealed that the pitting susceptibility of Al*_x_*CoCrFeNi-based HEAs increases with increasing Al content [19]. Kao et al. [20] also observed that Al*_x_*CoCrFeNi HEA shows high corrosion resistance in a 0.5M H_2_SO_4_ solution; however, its corrosion resistance decreases with increasing aluminum concentration to *x* = 1.0, due to the formation of a porous film that does not protect the surface against the aggressive ions.

Further research to improve the corrosion resistance of Al*_x_*CoCrFeNi-based HEAs focused on the effect of additionally introduced elements [16,17]. Copper is reported to be a promising alloying element to improve the corrosion resistance of Al*_x_*CoCrFeNi-based HEAs [21,22,23,24,25]. Lee et al. [21] described the positive effect of introduced copper on the corrosion resistance of the Al*_x_*CoCrFeNi-based HEA in 1N H_2_SO_4_ solution. Qiu [22] observed that Cu also had a positive effect on the corrosion resistance of AlCrFeNiCoCu HEA in 1M NaCl solution (*I*_corr_ = 3.23 × 10^−9^ A/cm^2^). Potentiodynamic polarization tests in 0.5M NaCl and 0.5M H_2_SO_4_ solutions indicated a high corrosion resistance for a series of FeCoNiCrCu_0.5_Al*_x_* alloys [23]. A significant decrease in pitting susceptibility of the AlFeNiCoCuCr alloy after heat treatment has been noted by Zhang et al. [24]. Liu et al. [25] also observed the positive effect of Cu on the corrosion behavior of a number of Al*_x_*CoCrCuFeNi HEAs, not only in salt but also in alkaline solutions. On the contrary, Pratskova et al. [26] noted that an excess concentration of copper adversely affects the corrosion resistance of Al_0.25_CoCrFeNiCu HEA in H_2_SO_4_ solution, since copper-rich areas are most exposed to sulfuric acid. Thus, more research is still required to analyze the effect of copper on the corrosion resistance of Al*_x_*CoCrFeNi-based HEAs.

On the other hand, it has been also reported that Cu-containing HEAs exhibit lower hardness and strength compared to Cu-free counterparts [8]. Hsu et al. [27] argued that the mechanical characteristics of such HEAs could be improved by introducing silver. The passivation of noble metals, in particular silver, in aggressive (sulfate) solutions has been fairly well studied [28]. It should be noted that the corrosion behavior of Al*_x_*CoCrFeNi HEAs alloyed with copper and silver has not yet been studied, which is the aim of the current study, and indicates its scientific novelty. The addition of silver, which is after hydrogen in the electrochemical series of voltages of metals, should, in our opinion, have a positive effect on the corrosion characteristics of high-entropy alloys in a sulfuric acid solution.

Therefore, this work studies the corrosion resistance of Al_0.5_CoCrFeNiCu*_x_*Ag*_y_* (*x* = 0.25, 0.5; *y* = 0, 0.1) HEAs in a sulfuric acid solution. This environment is equivalent to a medium with the simultaneous presence of SO_2_/SO_3_ vapor and water vapor/liquid, which is the typical environment in the extraction of natural resources such as oil and gas.

## 2. Materials and Methods

Al_0.5_CoCrFeNiCu*_x_*Ag*_y_* (*x* = 0, 0.25, 0.5; *y* = 0, 0.1) HEAs were prepared according to the procedure described in our previous work [29]. This composition was chosen based on previous findings [26], which indicated that in order to increase the corrosion resistance of copper-containing HEAs, it is necessary to suppress copper segregation and promote the complete alloying of copper into a multicomponent solid solution. To this end, in this study, the copper content in the composition of HEAs was reduced from *x* = 1 to 0.25–0.5. The aluminum content was also adjusted to 0.5, since many researchers have reported on the favorable effect of low aluminum content on the properties and processing of HEAs [19,20,21,23,25,30,31], probably due to the difficulties encountered in the melting of alloys with high aluminum content. For example, it is known that Al is easily oxidized in metal melts [32].

The preparation of working electrodes for the electrochemical research was described by us earlier in [26]. The composition of the samples and the exposed free surface areas of the working electrodes are presented in Table 1. Polarization measurements were carried out in a standard three-electrode YaSE-2 electrochemical cell (OJSC “Gomel Plant of Measuring Instruments”, Gomel, Belarus) with a platinum counter electrode using a P-30J potentiostat device (LLC “Elis”, Chernogolovka, Russia). The potentials were measured relative to a saturated silver chloride reference electrode EVL-1M3 (OJSC “Gomel Plant of Measuring Instruments”, Gomel, Belarus) at room temperature (25 °C) with a sweep rate of 5 mV/s. Thus, HEA samples were used as working electrodes, a silver chloride electrode served as the reference electrode, and a platinum electrode was the counter electrode. The corrosion tests were carried out in 0.5M H_2_SO_4_ solution. Before electrochemical tests, the surface of the electrodes was mechanically polished with emery paper, degreased with isopropyl alcohol, and then washed with distilled water.

The open circuit potential *E*_OCP_ was measured for 3600–4400 s in a cell filled with electrolyte solutions without applying current. This time was enough to stabilize the working electrodes. The obtained *E*_OCP_ of the samples are presented in Table 1. Corrosion parameters such as corrosion potential (*E*_corr_) and current density (*I*_corr_) were determined by the Tafel extrapolation method using both the cathodic and anodic branches of the polarization curves. A silver chloride electrode was used as the reference electrode, and then the potentials were recalculated to the scale of a normal hydrogen electrode.

The polarization resistance (*R*_p_) was calculated using the formula:(1)$$R_{p}=\frac{1}{2.303\left(\frac{1}{\beta_{a}}+\frac{1}{\beta_{c}}\right)I_{corr}}$$
where *β*_a_ and *β*_c_ are the slopes of the extrapolated anodic and cathodic straight lines of the Tafel equation, respectively.

The microstructure and surface morphology of the electrodes were examined using a JSM-7001F scanning electron microscope (SEM) (JEOL, Tokyo, Japan) equipped with an energy dispersive X-ray spectroscopy detector (EDS) (Oxford Instruments, Abingdon, UK) for quantitative chemical analysis. X-ray diffraction (XRD) was carried out on the thin sections of the samples on an Ultima IV diffractometer (Rigaku, Tokyo, Japan) using Cu Kα radiation.

## 3. Results and Discussion

### 3.1. Microstructure and Phase Composition of the As-Cast HEAs

Figure 1 presents the backscattered electrons SEM (BSE-SEM) micrographs and the corresponding EDS elemental mapping of the as-cast HEA samples. The EDS-determined chemical compositions are given in Table 2.

All samples were characterized by a dendritic microstructure, and in all cases the dendrites (D) were enriched in Co, Cr, and Fe. In the interdendritic (ID) regions, an area (ID1) enriched in Al, Ni, and Cu (if it is present in the sample) was distinguished for all the samples. When silver was added to the HEAs, a second Ag-rich phase (ID2) precipitated in the interdendritic regions. The segregation of copper into a separate ID3 phase was typical only for samples with a copper content of *x* = 0.5. The most complex microstructure was observed for the Al_0.5_CoCrFeNiCu_0.5_Ag_0.1_ HEA, in which all four described phases D, ID1, ID2, and ID3 were present.

Daoud et al. [33] observed a nearly similar microstructure for the Al_8_Co_17_Cr_17_Cu_8_Fe_17_Ni_33_ sample, in which copper segregated and Ni–Al–Cu-enriched regions coexist. Lin and Tsai [30] also noted the two-phase nature of the microstructure in the as-cast Al_0.5_CoCrFeNi sample, where the second phase enriched in Ni and Al. The precipitation of copper along the grain boundaries of the main matrix for a FeCoNiCrCu_0.5_Al_0.5_ HEA was described by Li et al. in [23]. The segregation of silver into a separate phase was described by Hsu et al. in [27]. Moreover, regions with coagulated silver could be observed in Al_0.5_CoCrFeNiCu_0.25_Ag_0.1_ and Al_0.5_CoCrFeNiCu_0.5_Ag_0.1_ samples (Figure 2). Similar coagulated silver areas have already been observed [27]. Detailed examination of these areas indicated eutectic characteristic of the Cu–Ag system [34]. With an increase in the concentration of copper in the samples, more copper was present in the silver-based globules.

For high-entropy alloys, the configurational entropy of mixing can be determined according to [35]:(2)$${∆S}_{mix}=-R\sum_{i=1}^{n}X_{i}\ln X_{i}$$
where *R* is the universal gas constant (*R* = 8.314 J∙mole^−1^∙K^−1^) and *X* is the atomic fraction of the elements. When Δ*S*_mix_ ≥ 1.5*R*, the alloy can be considered as a high-entropy system.

Using data from Table 2, Δ*S*_mix_ was calculated using the average composition and the composition of each of the microstructural components. The calculation results are given in Table 2. For Al_0.5_CoCrFeNi and Al_0.5_CoCrFeNiCu_0.25_, the entropy of mixing in each case exceeded 1.5*R*. For the remaining samples, according to the calculations for the average composition, they can be considered as HEAs. However, the obtained configurational entropy for each of the microstructural components, suggests that the samples are more likely a composite material, in which a Cu-rich medium-entropy phase and an Ag-rich low-entropy phase are distributed in a high-entropy matrix (two solid solutions D and ID1).

The XRD results of the as-cast HEAs are shown in Figure 3. All diffraction patterns show an fcc phase (D) and a bcc phase (ID1). For samples with silver, a second fcc (Ag-rich) phase (ID2) can be also detected.

The phase composition of Al*_x_*CoCrFeNi and Al*_x_*CoCrCu_0.5_FeNi HEAs is strongly affected by the aluminum concentration [23,36]; as the aluminum content increases from 0 to 1, the structure changes from fcc to bcc. According to Shi et al. [19], Kao et al. [20], and Abbaszadeh et al. [31], Al_0.5_CoCrFeNi HEA is characterized by a two-phase (fcc + bcc) structure, and the dendrites (D) can be attributed to the fcc solid solution, and the interdendritic phase (in our case, ID1) has a bcc crystal lattice. Thus, our data are consistent with those available in the literature.

### 3.2. Corrosion Behavior of HEAs in 0.5M H_2_SO_4_Solution

The polarization curves of the samples in the 0.5M H_2_SO_4_ solution are shown in Figure 4. The determined corrosion potentials (*E*_corr_), corrosion current density (*I*_corr_), polarization resistance (*R*_p_), range of passivation Δ*E*, and passivation current (*I*_pas_) are listed in Table 3.

The Al_0.5_CoCrFeNiCu_0.25_ sample showed the best corrosion current density and potential, where its corrosion resistance was better than Al_0.5_CoCrFeNi HEAs both in this work and elsewhere [20] (see Table 3). The Al_0.5_CoCrFeNiCu_0.5_ sample showed the worst corrosion behavior; however, all the presented samples showed better potentials and corrosion currents than stainless steel SS 304 [20,37]. The introduction of silver influenced the obtained values of *E*_corr_ and *I*_corr_. For the Al_0.5_CoCrFeNiCu_0.25_Ag_0.1_ HEA, the data was slightly worse than that of the sample without silver (see Table 3). For samples with a copper content of *x* = 0.5, the data on the silver-doped HEA, on the contrary, turned out to be significantly better than those of the sample without silver. In particular, the corrosion current decreased from 18.76 μA/cm^2^ for the Al_0.5_CoCrFeNiCu_0.5_ HEA to 7.05 μA/cm^2^ for the Al_0.5_CoCrFeNiCu_0.5_Ag_0.1_ HEA. It should also be noted that introducing silver expanded the range of passivation of the alloys. For samples with a copper content of *x* = 0.25, Δ*E* increased from 0.967 (for a sample without silver) to 1.282 (for the Ag-alloyed sample). Moreover, when the copper content was *x* = 0.5, by introducing silver into the composition of the alloy, the passivation region almost doubled from 0.458 to 0.745.

According to the polarization curves, silver-containing samples are characterized by not only passivation but also transpassivation, which is noted by additional peaks on the curves. Most likely, this can be associated with the presence of copper–silver eutectic zones and its interaction with the acid solution.

### 3.3. Microstructure and Chemical Composition after Corrosion Tests

Figure 5 and Figure 6 show the surface morphology and microstructure of the samples after corrosion tests. The compositions of the microstructural components are given in Table 4.

The Al_0.5_CoCrFeNi and Al_0.5_CoCrFeNiCu_0.25_ samples were characterized by an etching pattern with a clear separation of the second (bcc) phase, and it was noticeable that the surface of the sample without copper was pitted more strongly than that of the Cu-containing Al_0.5_CoCrFeNiCu_0.25_ HEA. The Al_0.5_CoCrFeNiCu_0.25_Ag_0.1_ sample exhibited a different behavior in the sulfuric acid solution: the dendrites and the interdendritic regions were practically not exposed to the aggressive medium (see Figure 5c), but etching traces appeared on the coagulated copper–silver eutectic areas, corresponding to the dissolution of copper (see Figure 6a,b). This is likely the reason for the slightly higher corrosion current density of the silver-containing sample compared to the silver-free sample (see Table 3). With an increase in the copper content to *x* = 0.5 and its segregation into a separate phase in the as-cast sample, a network of pores became visible in its microstructure after corrosion tests (see Figure 5d), which indicates the selective dissolution of copper in the sulfuric acid. This agrees with the data on corrosion potentials and currents, where the worst results were obtained for the Al_0.5_CoCrFeNiCu_0.5_ HEA. The introduction of silver had practically no effect on the copper dissolution. Silver is a stable material in acid solutions, therefore, in general, the Al_0.5_CoCrFeNiCu_0.5_Ag_0.1_ HEA showed slightly better results compared to the Al_0.5_CoCrFeNiCu_0.5_ sample (see Table 3).

As can be seen from Table 4, the interdendritic sections (ID1) were affected more severely by sulfuric acid compared to the dendrites (D). This is probably due to the lower chromium content of ID1, since Cr typically improves the corrosion resistance of solid solutions [18,26]. The interdendritic phase ID2 after corrosion tests was characterized by a lower concentration of copper, and ID3 (based on copper) completely dissolved, which is consistent with the observed microstructure.

According to studies [18,26], chromium-enriched regions act as a cathode, while depleted regions act as an anode and begin to be actively dissolved in an aggressive solution. The appearance of copper in the ID1 phase promoted the formation of a passivating film, so that copper additions had a positive effect until it began to segregate into a separate phase. After Cu segregation, these areas dissolved first, which negatively affected the corrosion characteristics of the HEAs. The alloying of silver contributed to the appearance of areas with copper–silver eutectic features in the HEA microstructure, which also led to the dissolution of copper from these areas. Therefore, among all the samples studied in this work, Al_0.5_CoCrFeNiCu_0.25_ HEA can be considered as the most promising corrosion-resistant alloy in the sulfuric acid solution. Further studies will be aimed at obtaining single-phase samples without the interboundary phase ID1. To this end, two strategies need to be implemented: (i) high-temperature annealing to suppress elemental segregation and (ii) tailoring the chemical composition to obtain a homogenous microstructure without pronounced segregations.

## 4. Conclusions

In summary, the corrosion behavior of Al_0.5_CoCrFeNiCu*_x_*Ag*_y_* (*x* = 0, 0.25, 0.5; *y* = 0, 0.1) HEAs in a 0.5M H_2_SO_4_ solution was studied. Among all the investigated alloys, Al_0.5_CoCrFeNiCu_0.25_ exhibited the best corrosion resistance in the sulfuric acid, which may be attributed to the lack of copper segregation into a separate phase. An increase in the copper content in the HEAs led to its precipitation along the grain boundaries; in this case, during corrosion tests, the dissolution of copper in the sulfuric acid solution and the appearance of a network of pores in the interdendritic space occurred. However, all samples showed better corrosion potentials and currents than SS 304 stainless steel. From the passivation point of view, the Al_0.5_CoCrFeNiCu_0.25_Ag_0.1_ HEA showed a rather wide range of passivation potentials (∆*E* = 1.282 V). The addition of silver to the HEAs was characterized by the appearance of passivation and repassivation peaks on the polarization curves, which may be ascribed to the presence of copper–silver eutectic zones in the microstructure of the samples and their interaction with the acid solution, in particular, the selective dissolution of copper. Further studies are ongoing aimed at obtaining Al_0.5_CoCrFeNiCu*_x_*Ag*_y_* samples without the precipitation of copper and second solid solution phases by maintaining the copper content at *x* = 0.25 and tailoring the aluminum concentration in the alloys.

## Figures and Tables

**Figure 1 materials-16-03585-f001:**
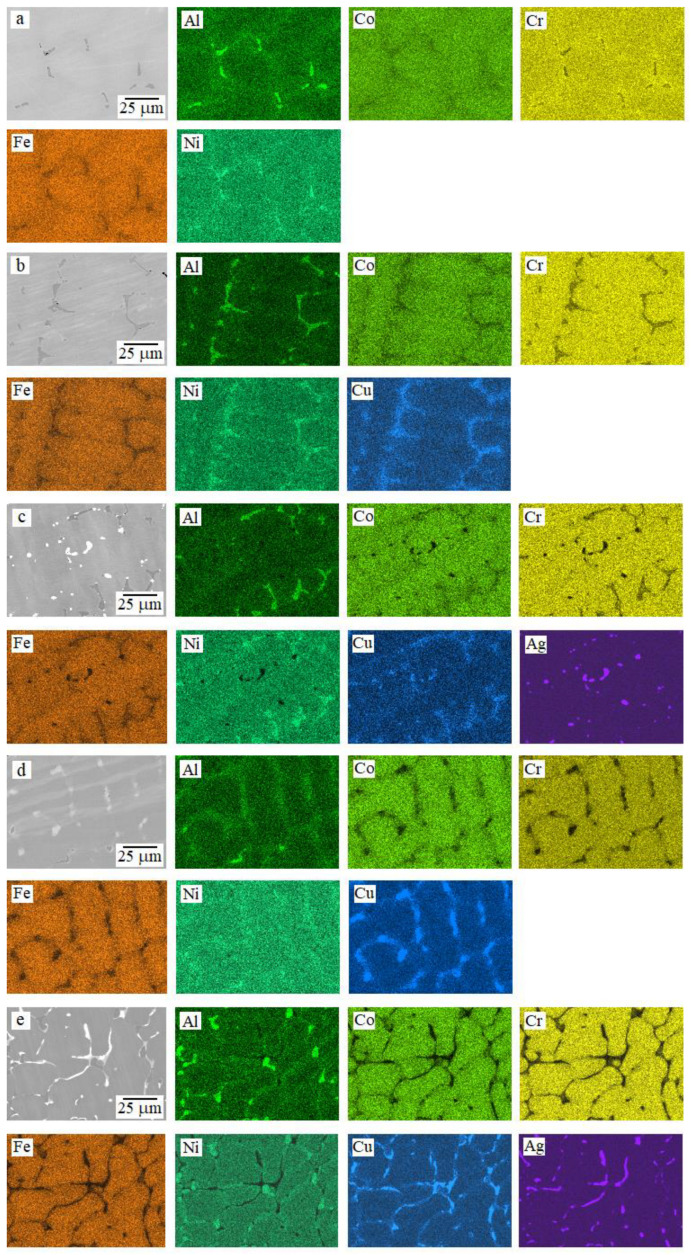
BSE-SEM images and the corresponding EDS mapping of the as-cast samples for (**a**) Al_0.5_CoCrFeNi, (**b**) Al_0.5_CoCrFeNiCu_0.25_, (**c**) Al_0.5_CoCrFeNiCu_0.25_Ag_0.1_, (**d**) Al_0.5_CoCrFeNiCu_0.5_, and (**e**) Al_0.5_CoCrFeNiCu_0.5_Ag_0.1_.

**Figure 2 materials-16-03585-f002:**
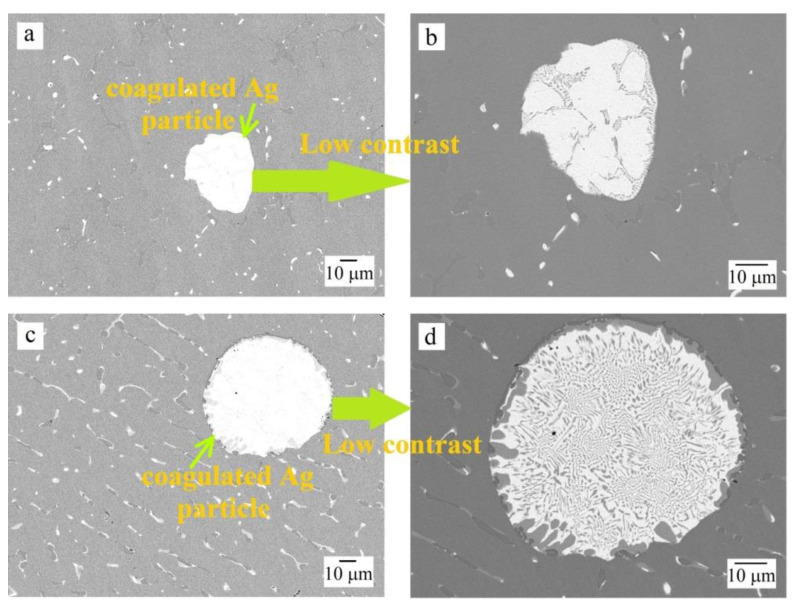
BSE-SEM images showing coagulated silver particle for (**a**,**b**) Al_0.5_CoCrFeNiCu_0.25_Ag_0.1_, (**c**,**d**) Al_0.5_CoCrFeNiCu_0.5_Ag_0.1_.

**Figure 3 materials-16-03585-f003:**
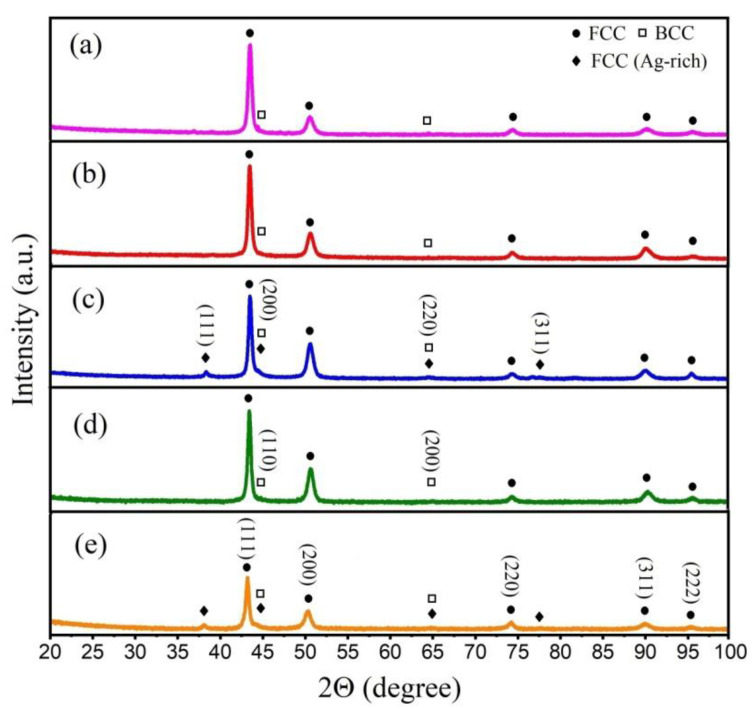
X-ray diffraction patterns for (**a**) Al_0.5_CoCrFeNi, (**b**) Al_0.5_CoCrFeNiCu_0.25_, (**c**) Al_0.5_CoCrFeNiCu_0.25_Ag_0.1_, (**d**) Al_0.5_CoCrFeNiCu_0.5_, and (**e**) Al_0.5_CoCrFeNiCu_0.5_Ag_0.1_.

**Figure 4 materials-16-03585-f004:**
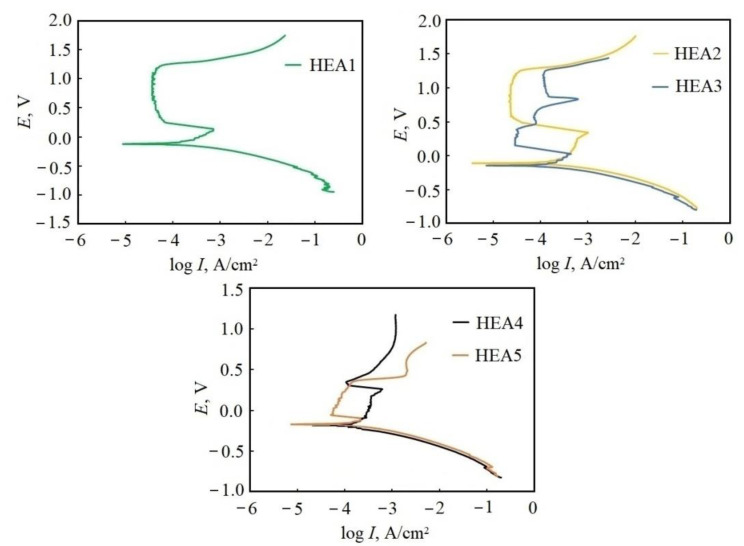
Polarization curves of HEAs samples in 0.5M H_2_SO_4_ solution. HEA1: Al_0.5_CoCrFeNi, HEA2: Al_0.5_CoCrFeNiCu_0.25_, HEA3: Al_0.5_CoCrFeNiCu_0.25_Ag_0.1_, HEA4: Al_0.5_CoCrFeNiCu_0.5_, HEA5: Al_0.5_CoCrFeNiCu_0.5_Ag_0.1_.

**Figure 5 materials-16-03585-f005:**
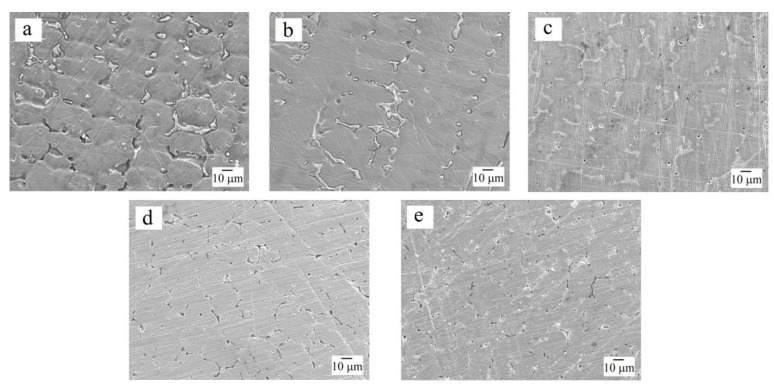
Secondary electron SEM images showing the surface morphology and microstructure of the samples after corrosion tests in 0.5M H_2_SO_4_ solution for (**a**) Al_0.5_CoCrFeNi, (**b**) Al_0.5_CoCrFeNiCu_0.25_, (**c**) Al_0.5_CoCrFeNiCu_0.25_Ag_0.1_, (**d**) Al_0.5_CoCrFeNiCu_0.5_, and (**e**) Al_0.5_CoCrFeNiCu_0.5_Ag_0.1_.

**Figure 6 materials-16-03585-f006:**
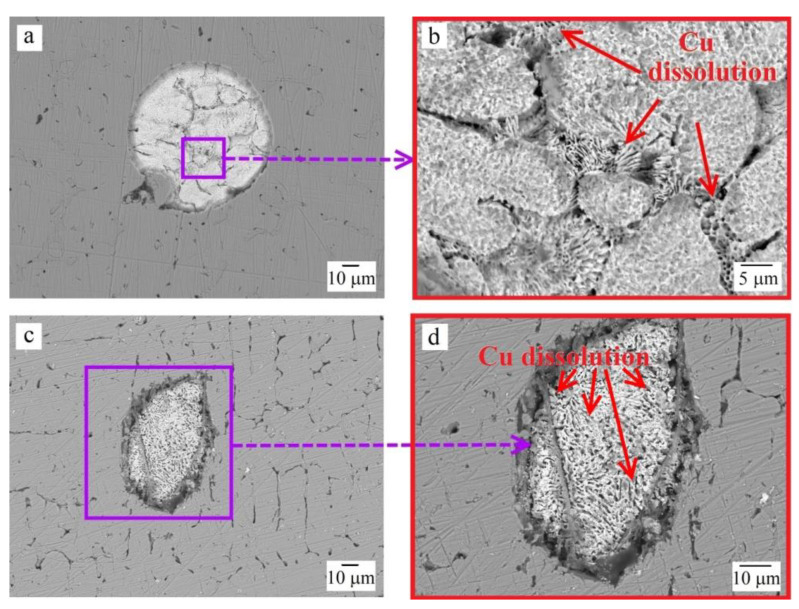
BSE-SEM images of the coagulated silver particle after corrosion tests: (**a**,**b**) Al_0.5_CoCrFeNiCu_0.25_Ag_0.1_, (**c**,**d**) Al_0.5_CoCrFeNiCu_0.5_Ag_0.1_.

**Table 1 materials-16-03585-t001:** The chemical formula (determined from EDS analysis), working surface area, and open circuit potentials (*E*_OCP_) for the studied samples.

HEA	Working Surface Area, mm^2^	*E*_OCP_, V (in 0.5M H_2_SO_4_)
Al_0.5_CoCrFeNi	10.5	+0.012
Al_0.5_CoCrFeNiCu_0.25_	10.5	+0.132
Al_0.5_CoCrFeNiCu_0.25_Ag_0.1_	10.5	+0.167
Al_0.5_CoCrFeNiCu_0.5_	10.5	+0.177
Al_0.5_CoCrFeNiCu_0.5_Ag_0.1_	10.5	+0.172

**Table 2 materials-16-03585-t002:** EDS chemical compositions (at %) and configurational entropy of mixing for microstructural components of the as-cast HEAs. Av—average composition; D—dendrite; ID—interdendritic areas.

Sample		Ag	Al	Co	Cr	Cu	Fe	Ni	Δ*S*_mix_
Al_0.5_CoCrFeNi	Av	–	10.76	22.62	22.51	–	21.24	22.87	1.58*R*
	D	–	8.57	23.65	23.81	–	22.96	21.01	1.56*R*
	ID1	–	26.75	15.14	16.57	–	11.02	30.52	1.54*R*
Al_0.5_CoCrFeNiCu_0.25_	Av	–	10.85	21.52	21.69	4.19	21.44	20.31	1.69*R*
	D	–	8.84	23.30	23.37	2.08	22.52	19.89	1.63*R*
	ID1	–	27.85	11.77	10.69	11.75	8.94	29.00	1.67*R*
Al_0.5_CoCrFeNiCu_0.25_Ag_0.1_	Av	1.95	10.09	20.94	21.04	4.23	20.52	21.23	1.75*R*
	D	–	8.17	23.03	23.89	2.31	22.52	20.08	1.63*R*
	ID1	–	26.40	11.69	12.41	10.65	10.53	28.32	1.69*R*
	ID2	80.22	–	3.58	–	13.11	–	3.09	0.67*R*
Al_0.5_CoCrFeNiCu_0.5_	Av	–	10.03	20.03	19.85	9.78	19.96	20.35	1.75*R*
	D	–	8.30	22.56	22.64	3.07	22.76	20.67	1.65*R*
	ID1	–	26.01	12.14	11.87	11.25	9.26	29.47	1.69*R*
	ID3	–	6.70	5.97	5.18	68.62	5.23	8.30	1.12*R*
Al_0.5_CoCrFeNiCu_0.5_Ag_0.1_	Av	1.90	9.55	19.72	19.77	9.75	19.57	19.74	1.81*R*
	D	–	8.27	22.91	22.23	4.74	22.95	18.90	1.68*R*
	ID1	–	27.49	11.74	7.89	13.79	8.36	30.73	1.65*R*
	ID2	83.67	–	–	–	14.32	–	2.01	0.51*R*
	ID3	2.31	6.83	3.66	1.80	72.64	2.50	10.26	1.02*R*

**Table 3 materials-16-03585-t003:** Potentials and corrosion current densities of alloys in 0.5M H_2_SO_4_ solution.

Alloy	*E*_corr_, V	*I*_corr_, μA/cm^2^	*R*_p_, kΩ cm^2^	*I*_pas_, μA/cm^2^	Δ*E*
Al_0.5_CoCrFeNi	−0.122 ± 0.002	6.05 ± 0.05	4.27 ± 0.05	36.3	1.144
Al_0.5_CoCrFeNiCu_0.25_	−0.112 ± 0.002	3.52 ± 0.02	7.81 ± 0.05	21.5	0.967
Al_0.5_CoCrFeNiCu_0.25_Ag_0.1_	−0.143 ± 0.002	5.21 ± 0.05	8.47 ± 0.05	28.8	1.282
Al_0.5_CoCrFeNiCu_0.5_	−0.200 ± 0.002	18.76 ± 0.05	2.29 ± 0.03	55.1	0.458
Al_0.5_CoCrFeNiCu_0.5_Ag_0.1_	−0.174 ± 0.002	7.05 ± 0.05	2.98 ± 0.03	43.6	0.745
Al_0.5_CoCrFeNi [20]	−0.084	13.4	–	6.4	1.200
SS 304 [20]	−0.185	45.3	–	19.1	0.750
FeCoNiCrCu_0.5_Al_0.5_ [23]	−0.112	4.190	–	–	0.550
SS 321 [23]	−0.003	2.248	–	–	0.325
Al_0.25_CoCrFeNiCu [26]	−0.164 ± 0.002	607 ± 3	0.088 ± 0.001	271.1	0.077
Al_0.3_CrFe_1.5_MnNi_0.5_ [37]	−0.194	2390	–	73.9	1.176
SS 304 [37]	−0.186	74.5	–	8.05	1.178

**Table 4 materials-16-03585-t004:** Compositions of microstructural components of HEA after corrosion tests according to EDS data (at. %). D—dendrite; ID—interdendritic space.

Sample		Ag	Al	Co	Cr	Cu	Fe	Ni	O	S
Al_0.5_CoCrFeNi	D	–	10.38	22.90	22.06	–	22.26	22.40	–	–
	ID1	–	30.40	13.42	13.13	–	10.07	25.46	7.07	0.45
Al_0.5_CoCrFeNiCu_0.25_	D	–	7.38	23.60	23.02	3.71	22.19	20.10	–	–
	ID1	–	27.25	13.01	8.96	8.43	8.24	31.29	2.43	0.39
Al_0.5_CoCrFeNiCu_0.25_Ag_0.1_	D	–	8.62	21.50	23.64	2.50	22.99	20.75	–	–
	ID1	–	22.11	13.82	14.25	6.84	11.01	30.63	1.14	0.20
	ID2	95.82	–	–	–	2.01	–	–	1.84	0.33
Al_0.5_CoCrFeNiCu_0.5_	D	–	9.81	20.93	22.15	6.08	19.76	21.27	–	–
	ID1	–	26.11	11.96	11.79	7.73	9.55	29.56	2.87	0.43
	ID3	Phase dissolved								
Al_0.5_CoCrFeNiCu_0.5_Ag_0.1_	D	–	7.48	22.71	23.98	5.25	19.44	21.14	–	–
	ID1	–	27.27	13.88	6.81	8.93	8.05	31.85	2.33	0.88
	ID2	96.60	–	–	–	2.38	–	–	0.80	0.22
	ID3	Phase dissolved								

## Data Availability

The data presented in this study are available on request from the corresponding author.

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
