# Peer review of "Corrosion Resistance of Al_0.5_CoCrFeNiCu*_x_*Ag*_y_* (*x* = 0.25, 0.5; *y* = 0, 0.1) High-Entropy Alloys in 0.5M H_2_SO_4_ Solution"

_materials, 2023, doi:10.3390/ma16093585_

Round 1

Reviewer 1 Report

1. The introduction section needs to explain the use/purpose of Al0.5CoCrFeNiCuxAgy, as well as the necessity and significance of studying this alloy. Why add silver to the alloy?

2. Silver only accounts for approximately 2% of the shares in Al0.5CoCrFeNiCu0.25Ag0.1 and Al0.5CoCrFeNiCu0.5Ag0.1. Please indicate the position of the coaged silver particles in Figure 2.

3. The XRD curve in Figure 3 should indicate the crystal plane index of each peak, similar to (111) (110), etc

4. There are too many overlapping curves in Figure 4, and from the graph, it can be seen that the corrosion potential of Al0.5CoCrFeNiCu0.5 is inconsistent with the potential shown in Table 4

5. In Figure 6b, there is a Cu solution.  In Figure 6d, the Cu content is higher than b. Please  supplement the details of Cu solution in Figure 6d. 

/

Author Response

Dear Reviewer, first of all we would like to thank you for your constructive comments, which helped us to improve the manuscript and to help us in our future work.

All corrections in the text of the manuscript are highlighted in yellow.

  1. The introduction section needs to explain the use/purpose of Al0.5CoCrFeNiCuxAgy, as well as the necessity and significance of studying this alloy. Why add silver to the alloy?

Answer: Corresponding corrections were made to the text of the manuscript.

“On the other hand, it has been also reported that Cu-containing HEAs exhibit lower hardness and strength compared to Cu-free counterparts [8]. Hsu et. al. [27] argued that the mechanical characteristics of such HEAs can be improved by introducing silver. The passivation of noble metals, in particular silver, in aggressive (sulfate) solutions has been fairly well studied [28]. It should be noted that the corrosion behavior of AlxCoCrFeNi HEAs alloyed with copper and silver has not yet been studied, which is the aim of the current study and indicates its scientific novelty. The addition of silver, which is after hydrogen in the electrochemical series of voltages of metals, should, in our opinion, have a positive effect on the corrosion characteristics of high-entropy alloys in sulfuric acid solution.”

  1. Silver only accounts for approximately 2% of the shares in Al0.5CoCrFeNiCu0.25Ag0.1 and Al0.5CoCrFeNiCu0.5Ag0.1. Please indicate the position of the coaged silver particles in Figure 2.

Answer: Corresponding corrections were made to the text of the manuscript.

  1. The XRD curve in Figure 3 should indicate the crystal plane index of each peak, similar to (111) (110), etc

Answer: Corresponding corrections were made to the text of the manuscript.

  1. There are too many overlapping curves in Figure 4, and from the graph, it can be seen that the corrosion potential of Al0.5CoCrFeNiCu0.5 is inconsistent with the potential shown in Table 4.

Answer: We rechecked the Figure 4, the data match the results presented in the Table 3.

  1. In Figure 6b, there is a Cu solution.  In Figure 6d, the Cu content is higher than b. Please  supplement the details of Cu solution in Figure 6d.

Answer: Corresponding corrections were made to the text of the manuscript.

Reviewer 2 Report

The reviewed work includes corrosion studies of a modified alloy belonging to the group of high entropy alloys (HEAs). The tests performed and the conclusions are clear.

Note that the electrochemical studies used the wrong reference electrode (saturated silver chloride) for a corrosive environment containing only sulfate ions. Generally, a reference electrode containing sulphate ions should be used in such conditions, e.g. a sulphate mercury reference electrode. The use of saturated silver chloride may cause chloride ions from the internal solution of the electrode to enter the corrosive environment (especially during longer measurements). This remark does not affect the obtained results, but it should be borne in mind in future research.

- Line 45 - AISI 1045 is not a stainless steel, it's a simple non-alloy steel

Author Response

Dear Reviewer, first of all we would like to thank you for your constructive comments, which helped us to improve the manuscript and to help us in our future work.

The reviewed work includes corrosion studies of a modified alloy belonging to the group of high entropy alloys (HEAs). The tests performed and the conclusions are clear.

Note that the electrochemical studies used the wrong reference electrode (saturated silver chloride) for a corrosive environment containing only sulfate ions. Generally, a reference electrode containing sulphate ions should be used in such conditions, e.g. a sulphate mercury reference electrode. The use of saturated silver chloride may cause chloride ions from the internal solution of the electrode to enter the corrosive environment (especially during longer measurements). This remark does not affect the obtained results, but it should be borne in mind in future research.

Answer: Thank you very much; we will take this comment into account in our future studies.

- Line 45 - AISI 1045 is not a stainless steel, it's a simple non-alloy steel

Answer: Corresponding corrections have been made to the text of the manuscript and highlighted with a green marker.

Round 2

Reviewer 1 Report

1. All the Figures in this paper are blurry, and the handwriting in the images is also blurry. It is necessary to be improved.

2. As can be seen from the paper, there are particles of various elements in the Al0.5CoCrFeNiCu0.25Ag0.1 and Al0.5CoCrFeNiCu0.5Ag0.1, such as Ag particles in  Figure 2  and/or Cu dissolutions in Figure 6.  So, it is not only necessary to indicate which crystal plane each peak on the XRD curve in Figure 3 belongs to, but also to indicate which metal element each peak belongs to. Please add the standard XRD curves of the corresponding element crystal in the figure for comparison. 

3. Figure 6b has a red box but Figure 6d does not. Please add a red box to Figure 6d. 

4. The references should be improved. 

5. There are too many overlapping curves in Figure 4. Please divide the overlapping curves into two figures so that readers can observe the details of the curves

1. All the Figures in the article are blurry, and the handwriting in the images is also blurry. It is necessary to be improved.

2. As can be seen from the paper, there are particles of various elements in the Al0.5CoCrFeNiCu0.25Ag0.1 and Al0.5CoCrFeNiCu0.5Ag0.1, such as Ag particles in  Figure 2 and/or Cu dissolutions in Figure 6.  So, it is not only necessary to indicate which crystal plane each peak on the XRD curve in Figure 3 belongs to, but also to indicate which metal element each peak belongs to. Please add the standard XRD curves of the corresponding element crystal in the figure for comparison. 

3. Figure 6b has a red box but Figure 6d does not. Please add a red box to Figure 6d. 

4. The references should be improved. 

5. There are too many overlapping curves in Figure 4. Please divide the overlapping curves into two figures so that readers can observe the details of the curves

Author Response

Dear Reviewer, first of all we would like to thank you for your constructive comments, which helped us to improve the manuscript and to help us in our future work.

All corrections in the text of the manuscript are highlighted in yellow.

1. All the Figures in this paper are blurry, and the handwriting in the images is also blurry. It is necessary to be improved.

Answer: All Figures are made with a resolution of at least 300 pixels per inch. It is possible that the apparent fuzziness is due to their small size, which is made according to the rules of the journal. For example, Fig. 2 in original dimensions looks like this.

2. As can be seen from the paper, there are particles of various elements in the Al0.5CoCrFeNiCu0.25Ag0.1 and Al0.5CoCrFeNiCu0.5Ag0.1, such as Ag particles in  Figure 2  and/or Cu dissolutions in Figure 6.  So, it is not only necessary to indicate which crystal plane each peak on the XRD curve in Figure 3 belongs to, but also to indicate which metal element each peak belongs to. Please add the standard XRD curves of the corresponding element crystal in the figure for comparison. 

Answer: We tried to plot such diagrams in Fig. 3, from this it became overloaded and unreadable. In addition, in the upper right corner of the Fig. 3, you can see the transcript for each peak.

3. Figure 6b has a red box but Figure 6d does not. Please add a red box to Figure 6d. 

Answer: Corresponding corrections were made to the text of the manuscript.

4. The references should be improved. 

Answer: Corresponding corrections were made to the text of the manuscript. The list of references in terms of the number of titles meets the requirements of the journal (at least 31 references) and is designed according to the rules of the MDPI.

5. There are too many overlapping curves in Figure 4. Please divide the overlapping curves into two figures so that readers can observe the details of the curves

Answer: Corresponding corrections were made to the text of the manuscript.
